# Determination of Aflatoxin M1 in Raw Milk from Different Provinces of Ecuador

**DOI:** 10.3390/toxins12080498

**Published:** 2020-08-03

**Authors:** Byron Puga-Torres, David Salazar, Mayra Cachiguango, Gabriela Cisneros, Carlos Gómez-Bravo

**Affiliations:** 1Milk Quality Control Laboratory, Faculty of Medicine Veterinary and Zootechnics, Central University of Ecuador, Jerónimo Leiton and Gilberto Gatto Sobral, 170521 Quito, Ecuador; dasalazarp@uce.edu.ec (D.S.); macachiguango@uce.edu.ec (M.C.); gaby_stefa18@hotmail.com (G.C.); 2Doctorate in Animal Science, Faculty of Zootechnics and Postgraduate School, La Molina National Agrarian University, La Molina Avenue, 15024 Lima, Peru; cagomez@lamolina.edu.pe

**Keywords:** Aflatoxin M1, raw milk, Ecuador, LFIA

## Abstract

Aflatoxin M1 (AFM1) is a mycotoxin from *Aspergillus flavus* and *A. parasiticus*, classified as carcinogenic and hepatotoxic. The objective of the present investigation was to determine its presence in raw milk from north-central Ecuador, constituted by the provinces of Pichincha, Manabí, and Santo Domingo de los Tsáchilas. These areas represent approximately 30% of Ecuadorian milk production. By the end of the investigation, a total of 209 raw milk samples were collected, obtained both during the dry (June and August) and rainy seasons (April and November) of 2019. AFM1 concentrations were measured with lateral flow immunochromatographic assays, and 100% of the samples were positive for this mycotoxin, presenting a mean value of 0.0774 μg/kg with a range of 0.023 to 0.751 μg/kg. These AFM1 levels exceeded the European Union regulatory limit of 0.05 μg/kg in 59.3% (124/209) of samples, while only 1.9% (4/209) exceeded the Ecuadorian legal limit of 0.5 μg/kg. By using non-parametric tests, significant differences were determined (*p* ≤ 0.05) between the provinces for months of study, climatic season (being higher in the dry season), and climatic region (greater in the coast region). On the other hand, there were no significant differences (*p* ≥ 0.05) between the types of producers or between production systems. Therefore, AFM1 contamination in raw milk does not present a serious public health problem in Ecuador, but a monitoring and surveillance program for this mycotoxin in milk should be developed to prevent consumer health problems.

## 1. Introduction

Aflatoxins (AF) are mainly produced by certain strains of *Aspergillus flavus* or *A. parasiticus* [1] and endanger public and animal health [2]. There are 18 different known AFs, and AFB1 is the most toxic and can contaminate various foods [3]. After ingestion, its high fat solubility favors gastrointestinal absorption and can reach the liver [4], where it is metabolized by the cytochrome P450 enzyme family and is hydrolyzed into Aflatoxin M1 (AFM1) or milk Aflatoxin [3]. It is then transferred to milk [5] and, thus, to milk derivatives and products [6].

AFB1 and AFM1 are carcinogenic and hepatotoxic [7]. The latter is the only mycotoxin with maximum residue limits (MRL) in milk [8], since it is not destroyed during the pasteurization process or during the preparation of dairy derivatives [9]. It has been verified that they can initiate and advance liver, lung, and colon cancer [10], reporting that 43.9% of the total annual cases of liver cancer in Bangladesh were associated with ingested AF [11].

AFM1 is found worldwide; however, its levels vary considerably in different regions, depending on the milk production system, climate, and the dairy species [12]. These problems are much more serious in low-income countries, where their presence can be common and even extreme [13], including contamination of breast milk [14] and infant formulas [15]. For this reason, the permanent study of AFM1 [16] is necessary to protect human health and trade [17].

The Ecuadorian Technical Standard (NTE) INEN 9 established a maximum limit of 0.5 μg/kg of AFM1 in milk, equal to the accepted value in the United States of America [18], but higher than the limit of 0.05 µg/kg in the European Union (EU). Since milk and its derivatives may be contaminated with AFM1 [19], and there are few investigations in Ecuador, the objective of the study was to determine the levels of AFM1 by using the lateral flow immunochromatographic assay (LFIA) with consideration for the major milk-producing provinces of Ecuador, the 2 climatic regions (inter-Andean and coast), seasons of the year (dry and rainy), type of producer (small, medium, and large), and by its production system (extensive, intensive, and mixed).

## 2. Results

The results obtained for provinces, months, climatic region, climatic season, type of producer, and production system are shown in Table 1 and Table 2, and Figure 1. The average AFM1 concentration was 0.077 μg/kg for milk samples from different Ecuadorian provinces. Milk analysis showed that the province of Manabí had an average (0.1256 μg/kg) of AFM1 approximately two times higher, than the average of the province of Pichincha (0.0639 μg/kg); curiously, the province of Santo Domingo had an average (0.0485 μg/kg) of approximately one third compared to Manabí, despite being neighboring provinces and sharing similar climatic characteristics (Table 1). The minimum value of 0.0230 μg/kg and the maximum value of 0.7510 μg/kg were observed in Pichincha. The level of AFM1 contamination in milk differed among the provinces (*p*-value of 5.332 × 10^−05^), and there were significant differences (*p* ≤ 0.05) between the results of the provinces during the study period, specifically between the provinces of Manabí and Pichincha and also between Manabí and Santo Domingo (Table 1). The results in the different climatic regions, indicated that the lowest mean was 0.064 μg/kg (inter-Andean region, Pichincha) and the highest was 0.1123 μg/kg (coast region, Manabí and Santo Domingo). A *p*-value of 0.0008957 (*p* ≤ 0.05) was obtained, and presented significant differences for the coastal region with the highest levels (Table 1). In the case of the Manabí province (located in the coast region), there is a constant temperature between 20 and 35 °C and high humidity between 82 and 90%, which are environmental conditions conducive to the growth of fungi and the production of mycotoxins.

The climatic season also differed in AFM1 contamination, where the highest levels were found in the dry season (mean of 0.0845 μg/kg and range was 0.0230 to 0.7510 μg/kg), while the lowest mean, being 0.0701 μg/kg, corresponded to the rainy season (Table 1). Thus, there were significant differences between the seasons during the study period (*p*-value of 0.005509).

For results from different sampling months (Table 2), the lowest mean was 0.0554 μg/kg (November, rainy season), and the highest was 0.1082 μg/kg (June, dry season), ranging from 0.0230 μg/kg (August) to 0.7510 μg/kg (June). Significant differences were observed between the study periods (*p*-value of 0.001406), specifically between April–June and June–November. It is noted that there were several higher and atypical values in some cases in relation to the other observations, especially for the month of June in the dry season (Figure 1). This was possibly the result of the rains being scarce during this period, causing a shortage in pasture material and, thus, a limitation for farmers feeding their cows with food such as silo (mainly corn), hay, and concentrated feed. These materials are more prone to *Aspergillus* contamination and, therefore, to the production of aflatoxins.

The results from different types of producers presented a minimum value of 0.0230 μg/kg and maximum of 0.7510 μg/kg., the highest mean (0.0817 μg/kg) was observed for the medium producers, while the lowest mean of 0.0577 μg/kg was from small producers. No significant differences were observed between the results for different types of producers (*p*-value of 0.5976) during the study period (Table 2). In the cases of medium producers with an extensive production system (grazing), the significant amount of outliers can possibly be attributed to the corresponding use of balanced feed at the time of milking.

The values obtained for different types of production systems (Table 2) presented a minimum value of 0.0230 μg/kg, maximum of 0.7510 μg/kg and the highest mean of 0.0837 μg/kg, all from systems of mixed production. On the other hand, the lowest mean of 0.0710 μg/kg was observed for the intensive system. There were no significant differences (*p*-value of 0.3493) between the production systems.

All raw milk samples from various parts of Ecuador were positive for AFM1, and this confirmed the ubiquitous presence of this mycotoxin, possibly due to widespread contamination of feeds and fodders with aflatoxins. A survey was carried out on the feed that the cows received from the farms, sampled in the previous week of milk collection, and found that approximately 99% of farms (207/209) fed their animals with fresh grass, 30.6% (64/209) with silage, 11.5% (24/209) with hay, and 90% (188/209) with balanced feed. The presence of AFB1 in the feed of dairy cows was not analyzed.

Only 1.9% of the samples (4/209) did not comply with the MRL of 0.5 μg/kg, established by the Ecuadorian regulations (Figure 1 and Table 3). Of these samples, 25% (1/4) pertained to Pichincha (inter-Andean region) and the other 75% (3/4) to Manabí (coast region); no sample over the standard value pertained to Santo Domingo de los Tsáchilas. When compared to European regulations, which allow a MRL of 0.05 μg/kg, 59.3% (124/209) of the samples do not comply with the above mentioned legislation (Figure 1 and Table 3). In this respect, 63.7% (79/124) of the samples over this limit came from Pichincha (inter-Andean region), 32.3% (40/124) from Manabí (Costa region), and 4.0% (5/124) from Santo Domingo de los Tsáchilas (coast region). Considering these same regulatory standards, 48.4% (60/124) of samples over the limit were collected during the rainy season (April 24 and November 36), and the remaining 51.6% (64/124) were from the dry season (38 in June and 26 in August). Likewise, 28.2% (35/124) pertained to large producers, 59.7% (74/124) to medium producers, and 12.1% (15/124) to small producers. In addition, 77.4% (96/124) have an extensive grazing system, 18.6% (23/124) carry out a mixed system (grazing and stabled), and 4.0% (5/124) are of an intensive system.

## 3. Discussion

Mycotoxins can cause a serious public and animal health problem, within which Aflatoxins can cause carcinogenic effects when ingested in food. In Ecuador, there are few studies of its presence in food, so the purpose of this research was to determine the levels of AFM1 in raw milk from 3 of the provinces with the largest contribution to total milk production in Ecuador.

Developing countries—such as Ecuador—could have high levels of AFM1 due to lack of producer knowledge, technology, and poor facilities, all of which contribute to the growth of toxin-producing fungi [20]. For that reason many countries have established different maximum limits for AFM1 in raw milk (Table 3); for example, the EU and some other countries use 0.05 ug/kg, while the United States, Latin-American (except Chile), and some Asian countries use 0.5 ug/kg.

Our results highlight the prevalence of AFM1 in 100% of raw milk samples from the provinces that represent more than 30% of the total Ecuadorian production [21]. There was an average of 0.0774 μg/kg, and these were higher than those obtained in 5 provinces of the Ecuadorian inter-Andean region, where they observed an average of 0.034 µg/kg (range 0.019–0.07 µg / kg) in 50 milk samples from collection centers (small producers). In this same study, 100% of the samples analyzed were below the MRL set by Ecuadorian regulations, and only 4% (2/50) of the samples exceeded the European regulations [22]. Likewise, the present data totally contradicts the study carried out in the Biblán-Azuay-Ecuador canton, where the presence of AFM1 was not found in any of the 88 samples from 22 farms of medium-sized producers [23].

According to Table 3, the present results also contradict those of other Latin American countries, whose average was 0.037 µg/kg [24]. For milk from Arequipa-Peru, no milk samples were found to contain AFM1 [25]. In Argentina, the AFM1 concentration was 0.059 µg/kg, and that milk exceeded the MERCOSUR MRL (0.5 µg/kg) and the EU regulations by 0.81% and 32.65%, respectively [26]. In El Llano-Mexico, 54.16% of 216 samples presented mycotoxins, but none exceeded the Mexican MRL (0.5 μg/kg), and 27.31% exceeded the European MRL [27]. In the same country, infant formulas were analyzed, and the researchers found that 20% of 55 samples had AFM1 values above the EU MRL of 0.25 μg/kg [28]. Likewise, in Valparaíso-Chile, 36% of 44 samples analyzed exceeded European regulations [29].

In developed countries, the presence of AFM1 has been controlled (Table 3). For example, between 2013 and 2018, 31,702 raw milk samples in Italy (556,413 tons) presented a monthly average of AFM1 between 0.00719 and 0.02253 μg/kg [30]. In China, 75.2% of 133 raw milk samples were AFM1 positive (mean of 0.0159 μg/kg), but none exceeded the Chinese or European MRL [31]. However, 4.64% of 1207 (56/1207) samples of milk powder for infants were positive for AFM1, but none exceeded the Chinese MRL either [32]. In developing countries, on the other hand, the problem of AFM1 is variable (Table 3). For example, an overall average of 0.056 μg/kg was reported in Iran, according to a meta-analysis study [33]. In India, 20.67% of 150 samples contained levels above the MRL of local legislation, being 0.5 μg/kg [34]. In Punjab-Pakistan, 70% of 690 raw milk samples were above the MRL of 0.5 μg/kg [35]. In Canakkale-Turkia, 3.3% of 120 samples exceeded local limits [36]. Likewise, all 175 milk samples from Jordan contained AFM1, and 66% did not comply with European legislation and 23% with American MRL [37]. In Nairobi-Kenya, all 96 raw milk samples contained AFM1, and 66.6% were above the EU MRL and 7.5% above the local regulations (0.5 μg/kg) [38].

**Table 3 toxins-12-00498-t003:** AFM1 maximum limits in different countries and raw milk contents from various research.

Country	AFM1 Maximum Limits in Raw Milk (µg/kg)	Number of Samples	Meanμg/kg	Positive Samples to AFM1	Samples > 0.5 μg/kg	Samples > 0.05 μg/kg	Type of Analysis	Detection Limit (μg/kg)	Reference
USA	0.5								
EU	0.05								
Ecuador	0.5	209	0.077	209 (100%)	4 (1.91%)	124 (59.3%)	LFIA	0.025	This study
		50	0.034	50 (100%)	0 (0%)	2 (4%)	LFIA	0.02	[22]
		88	0 (0%)	0 (0%)	0 (0%)	0 (0%)	ELISA/SNAP	0.05	[23]
Peru	0.5	40	0 (0%)	0 (0%)	0 (0%)	0 (0%)	ELISA/SNAP	0.05	[25]
Argentina	0.5	-	0.059	100%	(0.81%)	(32.65%)	-	-	[26]
Mexico	0.5	216	0.026	117 (54.16%)	0 (0%)	31 (27.31%)	ELISA	0.05	[27]
Chile	0.05	44	0.06	33 (75%)	33 (75%)	16 (36.4%)	HPLC-FL	0.0091	[29]
Italy	0.05	31702	0.00719–0.02253	-	-	-	LFIA and ELISA	0.025	[30]
China	0.5	133	0.0159	100 (75.2%)	0 (0%)	0 (0%)	LC-MS/MS	0.005	[31]
Iran	0.1	Meta-analysis 77 studies	0.056	Meta-analysis					[33]
India	0.5	150	0.262	77 (51.33%)	31 (20.67%)	46 (30.66%)	HPLC	0.052	[34]
Pakistan	0.5	690	0.640	690 (100%)	483 (70%)	690 (100%)	ELISA	0.1	[35]
Turkey	0.5	120	0.0051	107 (89.2%)	4 (3.33%)	13 (10.83%)	ELISA	0.005	[36]
Jordan	0.5	175	0.0689	175 (100%)	40 (23%)	115 (66%)	ELISA	0.025	[37]
Kenya	0.5	96	0.2903	96 (100%)	7 (7.5%)	64 (66.6%)	ELISA	0.005	[38]

LFIA: lateral flow immunochromatographic assays; ELISA/SNAP: Enzyme-Linked ImmunoSorbent Assay; SNAP: Software Non-functional Assessment Process; HPLC: High-performance liquid chromatography; HPLC-FL: High-performance liquid chromatography with fluorescence detector; LC-MS/MS: Liquid chromatography coupled with tandem mass spectrometry.

Regarding the climatic season, some authors have indicated that it does not influence the presence of AFM1 [24]; however, differences were found between the seasons in this study (*p*-value of 0.005509), where the highest values were observed for the dry season. This finding corroborates the study carried out on 48 milk samples from Chiapas-Mexico, where averages of 0.55 μg/kg were obtained in the dry season and 0.13 μg/kg in the rainy season. This observation was attributed to feeding animals with contaminated silage and feed [39], as heat stress and drought can cause increased concentrations of AFM1 in milk [40]. However, other authors have indicated that the highest concentrations occur in winter, as the temperature and humidity could contribute to the production of AFB1 in the harvested or stored food [41,42]. Moreover 100% of samples analyzed in Iran during the winter had the highest concentrations of AFM1 [43], and in a meta-analysis study of 77 cases, it was also determined that the highest levels were in winter [33]. This conclusion is also corroborated by a study in Punjab-Pakistan, where the highest mean (0.875 μg/kg) was found in winter [35].

Regarding type of producers, no significant differences were observed (*p*-value of 0.5976); the highest mean was found in medium-sized producers, what contradicts the results obtained in Iran, where the highest averages were found in small producer farms [44], and in Serbia, the means of small producers (0.23 μg/kg) were also much higher than that of large producers (0.0079 μg/kg) [45]. Regarding the types of production systems, there were no significant differences (*p*-value of 0.3493); the highest average was observed in mixed production systems and the lowest in intensive systems. This contradicts values reported in Kenya, where food tends to be contaminated by aflatoxins in intensive systems [46] with corn being the most prone to contamination, in addition to other diets destined for dairy cattle, such as corn silage, rice hay, and alfalfa hay [47].

## 4. Conclusions

The present study contributes important results, obtained from three provinces of Ecuador, representing more than 30% of the total milk raw production. The samples were collected during two seasons, being two months categorized by climatic season: April and November (rainy season) and June and August (dry season). This study confirms the presence of AFM1 in raw milk from different provinces of Ecuador, possibly due to contamination of feeds and fodders with aflatoxins, and these results suggest an increase compared to previous studies conducted in Ecuador. The highest mean of AFM1 was found in the province of Manabí (coast region), possibly due to the high temperature and humidity of the area, characteristics that favor the growth of fungi. Likewise, in the dry season, the lack of rain causes a shortage of pasture, so the farmers feed their cows with silo, (mainly corn), hay and concentrated feed, which are more prone to *Aspergillus* contamination. The levels of AFM1 are above the maximum limit allowed by the European Union (0.05 μg/kg) for 59.3% of the samples. However, only 1.9% of the analyzed milk samples had levels above the permissible limit by Ecuadorian and American legislation (0.5 μg/kg). Therefore, AFM1 contamination in raw milk does not represent a serious public health problem in Ecuador, but it is necessary to continuously monitor its milk and make efforts to reduce the AFB1 content in the diet of dairy cattle and, therefore, AFM1 in raw milk. This can be achieved by increasing controls and educating dairy farmers on public health problems caused by this mycotoxin, especially in Manabí province and in the dry season.

## 5. Materials and Methods

### 5.1. Sample Collection

Ecuador produced a total of 5.02 million liters of milk per day in 2018, and the per capita consumption was between 90–95 kg/person/year. The Pichincha province (in the inter-Andean region) was the largest producer with approximately 16% of total Ecuadorian milk production, followed by the province of Manabí (coast region) with about 12% and the province of Santo Domingo de los Tsáchilas (coast region) with 4% [21]. The Sierra or inter-Andean region has a cold temperate climate throughout the year, while in the coastal region, there is always a warm climate. There is only a rainy period between November to May and a dry period between June and September [48].

The number of samples depends on the production volume of each province, which is greater in Pichincha, followed by Manabí and then Santo Domingo de los Tsáchilas. A total of 209 samples were collected between the months of April and November (rainy season) and June and August (dry season) in 2019. Of the samples collected, 72.3% pertained to the province of Pichincha (151/209), 22.9% to Manabí (48/209), and 4.8% to Santo Domingo de los Tsáchilas (10/209). Regarding the climatic season, 53.1% corresponded to the rainy season (111/209) and 46.9% to the dry season (98/209). In April, June, and August, 49 samples (23.5%) were taken each month, while the remaining 29.7% (62/209) of the samples were collected in November.

Raw milk samples (*n* = 209) were collected from the central-northern region of Ecuador (Figure 2). Among these, 151 samples (72.3%) were from the Pichincha province (temperature between 5–20 °C, 70–75% humidity, and 2500–3200 m above sea level), 48 (22.9%) from Manabí (temperature between 20–35 °C, 82–90% humidity, and 150–300 m above sea level), and 10 (4.8%) from Santo Domingo de los Tsáchilas (temperature between 21–32 °C, 80–90% humidity, and 150–300 m above sea level). One hundred and eleven samples (53.1%) were collected during the rainy season (April and November) and 98 samples (46.9%) in the dry season (June and August) of 2019.

The samples were from small (26/209), medium (127/209) and large scale (56/209) dairy farms, which means they possess from 1 to 20, from 21 to 100, and more than 100 cows in production, respectively, based on its production system (extensive, intensive, or mixed). They were collected directly from cooling tanks of each farm, following the milk sampling guidelines indicated by NTE INEN-ISO 707 [49]. They were later refrigerated and transported using coolers at a temperature between 2 and 5 °C to the Milk Quality Control Laboratory of the Faculty of Veterinary Medicine and Zootechnics of the Central University of Ecuador, where they were stored at −20 °C until the respective analysis.

### 5.2. Analysis of AFM1 in Milk by LFIA

Before analyzing the samples, the milk was thawed at room temperature and subsequently centrifuged at 4000× *g* for a period of 10 min. Afterwards, the fat layer was removed from the top of the milk.

AFLA M1-V VICAM^®^ analysis tests (Vicam, 34 Maple Stret, Milford, MA 01757, USA) were used for detection of specific monoclonal antibodies for AFM1. Samples were incubated in a Delvotest^®^ Incubator DSM-MiniS-11548 (Nangzhou Allsheng Instrument Co., Ltd., Building 1 & 2, Zheheng Science Park, Zhuantang Town, Xihu District, Hangzhou, China), and reading was performed using a Vertu 1648 Lateral Flow Reader (Vicam, 34 Maple Stret, Milford, MA, USA), whose optical detector is oriented along the test strip and transforms the data to a mycotoxin concentration through a calibration curve.

The kit uses monoclonal antibodies to accurately detect and measure AFM1 with lower and upper detection limits of 0.025 μg/kg (25 ppt) and 0.75 μg/kg (750 ppt), respectively. The process eliminates the need for hazardous solvents by using a water based undiluted extraction procedure.

The milk sample (200 uL) was added to the AFLAM1-V strip test vial containing the conjugate. It was then mixed 3 times for 5 s each and incubated at 40 °C for 10 min with the test strip. The strip was then inserted into the Vertu reader for measurement, and the result appeared directly on the digital screen for recording.

### 5.3. Statistical Analysis

All results are expressed as mean, minimum, and maximum concentrations of AFM1. Before the analysis, the Shapiro–Wilk test was performed, obtaining a *p*-value of 2.2 × 10^−16^ (*p* ≤ 0.05). Because a normal distribution of the data was not observed, non-parametric tests were additionally used [50]. The Kruskal Wallis test was used to compare the results of AFM1 in raw milk by province, type of producer, production system, and sampling months. In the case of significant differences, a post-hoc analysis was performed using the Mann–Whitney test with a Bonferroni correction to decrease the probability of type I error. The Wilconox test also was used to compare the values corresponding to the season and climatic region. Statistical software RStudio version 1.2.5019 (RStudio Inc. Boston, MA, USA) was used with a significance of *p* < 0.05 for all analyses in order to determine whether any of the differences were statistically significant.

## Figures and Tables

**Figure 1 toxins-12-00498-f001:**
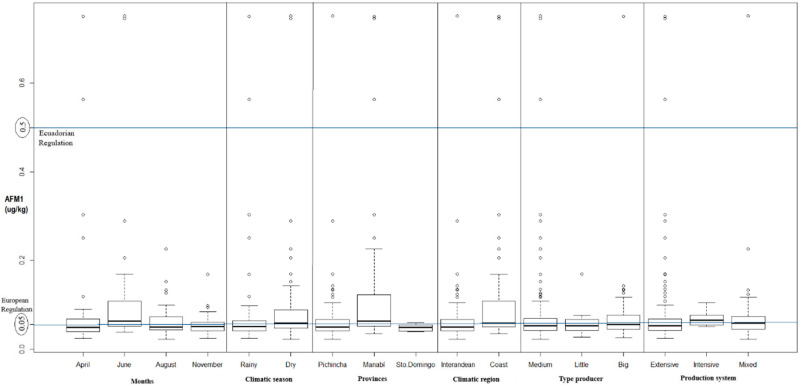
Boxplot with the values by months, climatic season, provinces, climatic region, producer type, and production system.

**Figure 2 toxins-12-00498-f002:**
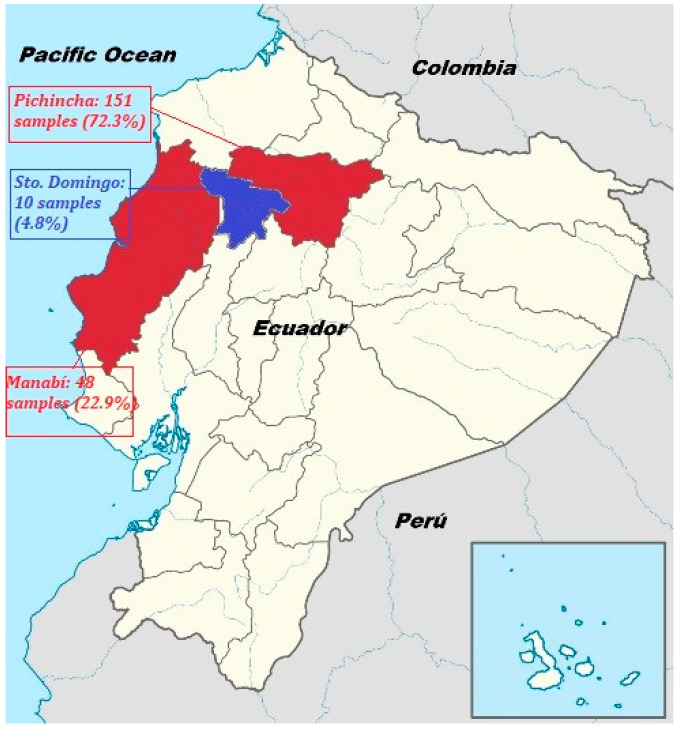
Map of Ecuador marked by provinces where the sampling was carried out.

**Table 1 toxins-12-00498-t001:** Aflatoxin M1 (AFM1) minimum, median, mean, and maximum values obtained by province, climatic region, and climatic season.

Variable	Minimum (μg/kg)	Median (μg/kg)	Mean (μg/kg)	Maximum (μg/kg)	*p*-Value
Province					
Manabí	0.0350	0.0640	0.1256	0.7500	5.332 × 10^−05^ (*p* ≤ 0.05)
Pichincha	0.0230	0.0510	0.0639	0.7510	
Santo Domingo de los Tsáchilas	0.0400	0.0500	0.0485	0.0600	
Climatic season					
Dry	0.0230	0.0580	0.0845	0.7510	0.005509 (*p* ≤ 0.05)
Rainy	0.0250	0.0520	0.0701	0.7500	
Climatic region					
Coast	0.0350	0.0590	0.1123	0.7500	0.0008957 (*p* ≤ 0.05)
Inter-Andean	0.0230	0.0510	0.0640	0.7510	

**Table 2 toxins-12-00498-t002:** AFM1 minimum, median, mean and maximum values obtained by months, producer type, and production system.

Variable	Minimum (μg/kg)	Median (μg/kg)	Mean (μg/kg)	Maximum (μg/kg)	*p*-Value
Month					
June	0.0390	0.0640	0.1082	0.7510	0.001406 (*p* ≤ 0.05)
April	0.0250	0.0500	0.0862	0.7500	
November	0.0250	0.0520	0.0554	0.1690	
August	0.0230	0.0510	0.0656	0.2260	
Producer Type					
Large	0.0270	0.0565	0.0768	0.7500	0.5976 (*p* ≥ 0.05)
Medium	0.0230	0.0540	0.0817	0.7510	
Small	0.0280	0.0540	0.0577	0.1700	
Production System					
Intensive	0.0520	0.0660	0.0710	0.1050	0.3493 (*p* ≥ 0.05)
Mixed	0.0230	0.0590	0.0837	0.7510	
Extensive	0.0250	0.0540	0.0761	0.7500	

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
