# Peer review of "Determination of Aflatoxin M1 in Raw Milk from Different Provinces of Ecuador"

_toxins, 2020, doi:10.3390/toxins12080498_

Round 1

Reviewer 1 Report

Dear Editor,

The objective of this work is the determination of the presence of Aflatoxin M1 (AFM1) in raw milk from north-central Ecuador, AMF1 is a mycotoxin from Aspergillus flavus and A. parasiticus, classified as a carcinogenic and hepatotoxic. Authors presented results about the presence of this micotoxin. These data are interesting, I do consider that experimental results shown are enough for publishing this paper. Experimental data regarding important results of the presence of AFM1 in raw milk of Ecuador is confirmed.

Moreover, the paper is well structured. Therefore, I do recommend considering the publication of this paper in the present version, and I suggest to the Authors go to the TV and the papers with this important result.

Sincerely,

                  The reviewer

Author Response

Response to Reviewer 1 Comments

Point 1: English language and style: “English language and style are fine/minor spell check required”

Response 1: We have improved writing in English with the help of a native speaker of the language.

Byron Puga Torres

Reviewer 2 Report

The authors performer a survey in Ecuador on AFLM1 contamination in milk samples, evaluating several factors that could influence mycotoxin levels (seasonality, geographic area, type of farming [intensive and extensive], farm size). The results could be very interesting, considering that no similar studies are available in Ecuador, but the manuscript needs major revisions essentially in the results and discussions section, and a thorough revision of the English language to be accepted by Toxins.

Major revisions

Results section

- The Authors should rewrite the results,describing before the results in relation to each factor considered (from line 66 to line 94) then the general conclusions (from line 54 to line 65).

- The Authors should modifiy Table 1 and 2 in the following way:

  1. a) with which values do you compare and get those p values?
  2. b) It is possible to make a rank among the results on provinces or months or type producers or production systems?
  3. c) Having made a non-parametric analysis, it is better to put the values of median instead of mean.

- The authors should explain and comment the data showed in the figure 1  dot-box graph, i.e the different  variability of the results (especially in June, in the province of Manabi and in the dry season), and  the different number of outliers in relation to the different MLR values (European or Ecuadors) that allow to select some more sensitive factors (e.g. extensive> medium producer> interandean> Pichinca).

Discussion section

- The discrepancy between the data obtained in the study and those of previous works is very confusing because sometimes the authors consider  for the comparison the incidence of positive cases, sometimes the levels of AFLM1 found or some parameters considered (e.g. farm size or seasonality or type of farming). The text is very confusing. The paper could  be clearer if bibliographic data, considered for the comparison, were reported in a table with all useful information (e.g. number of samples, chemical analysis used, detection limit, factor to consider).

. A possible suggestion could be to include a table in which to report for each contry considered for the comparison the MLRs and the incidence of samples exceeding these limits, as reported in the reference of Guo et al (2019), in order to understand the real risk for health in Ecuador and compare it with that of other countries.

- Another problem is the lack of availability or accessibility of many articles considered because they are written in Spanish or not available. This problem further complicates the understanding of the discussion and does not allow an accurate comparison. Also for this reason, the inclusion of the table would allow to make it more comprehensible and clear

Minor revisions:

line 47……because milk and its derivatives may be exposed to AFLM1. The Authors should modify the sentence

The numbering of the figures is not in the right order in the text (line  59 figure 2;  line 66 figure 1)

Lines  146-147…. This  contradicts what is indicated in Kenya, where food tends to be contaminated by FA,… The authors should explain the meaning of abbreviation  FA

Lines 206-208: The authors should check the detection limit and the limit of quantification of the analytical method used in this study. They are not correct.

Letter to editor

The study is modest in its  content and structure but  it provides results for a country not easily available. It lacks a substantial bibliographic and informative framework on the problem of AFLM1 cotamination in milk, but it contains some bibliography that, if organized and adeaquatly shown, could be usefull. From their results  resulted that  the real risk of milk is very low, having only 2% of the samples with limits higher than the American MLR, values that reach 60% considering the European limits. The authors show some figures that are not commented and from which  it could select  some sensitive factors involved in  AFLM1 contamination of milk. The results showed in this study seem to be different from those of other Latin American countries but it is not clear the reason and the authors did not suppose any hypothesis. Having considered several variables that may affect AFLM levels in milk,  the authors did not  make any conclusions

Author Response

Response to Reviewer 2 Comments

Point 1: English language and style “Moderate English changes required”

Response 1: We have improved writing in English with the help of a native speaker of the language.

Point 2: Results section: The Authors should rewrite the results, describing before the results in relation to each factor considered (from line 66 to line 94) then the general conclusions (from line 54 to line 65).

Response 2: The results section was restructured and improved.

Point 3: Results section: The Authors should modifiy Table 1 and 2 in the following way:

  1. a) with which values do you compare and get those p values?
  2. b) It is possible to make a rank among the results on provinces or months or type producers or production systems?
  3. c) Having made a non-parametric analysis, it is better to put the values of median instead of mean.

Response 3: Table 1 and Table 2 were corrected at the following points: there was indicated where the p value was obtained; - a ranking was made based on the average, from the highest value to the lowest; - median values were included.

Point 4: Results section: The authors should explain and comment the data showed in the figure 1  dot-box graph, i.e the different  variability of the results (especially in June, in the province of Manabi and in the dry season), and  the different number of outliers in relation to the different MLR values (European or Ecuadors) that allow to select some more sensitive factors (e.g. extensive> medium producer> interandean> Pichinca).

Response 4: The explanation of Figure 1 was add, in the last paragraph of the results section

Point 5: Discussion section: The discrepancy between the data obtained in the study and those of previous works is very confusing because sometimes the authors consider  for the comparison the incidence of positive cases, sometimes the levels of AFLM1 found or some parameters considered (e.g. farm size or seasonality or type of farming). The text is very confusing. The paper could  be clearer if bibliographic data, considered for the comparison, were reported in a table with all useful information (e.g. number of samples, chemical analysis used, detection limit, factor to consider).

Response 5: Table 3 was added, where the requested data is indicated.

Point 6: Discussion section: A possible suggestion could be to include a table in which to report for each contry considered for the comparison the MLRs and the incidence of samples exceeding these limits, as reported in the reference of Guo et al (2019), in order to understand the real risk for health in Ecuador and compare it with that of other countries.

Response 6: Table 3 was added, where the requested data is indicated.

Point 7: Discussion section: Another problem is the lack of availability or accessibility of many articles considered because they are written in Spanish or not available. This problem further complicates the understanding of the discussion and does not allow an accurate comparison. Also for this reason, the inclusion of the table would allow to make it more comprehensible and clear

Response 7: Some articles are in Spanish because they were made in Latin America. Regarding the bibliography, it has been reviewed and corrected (the links).

Point 8: Line 47……because milk and its derivatives may be exposed to AFLM1. The Authors should modify the sentence

Response 8: Corrected in Line 59 and 60 "Since milk and its derivatives may be contaminated with AFM1".

Point 9: The numbering of the figures is not in the right order in the text (line  59 figure 2;  line 66 figure 1

Response 9: Corrected in Line 81, 126, 130 and 139.

Point 10: Lines  146-147…. This  contradicts what is indicated in Kenya, where food tends to be contaminated by FA,… The authors should explain the meaning of abbreviation  FA

Response 10: Corrected in Line 216: “Aflatoxins”

Point 11: Lines 206-208: The authors should check the detection limit and the limit of quantification of the analytical method used in this study. They are not correct

Response 11: Corrected in Line 288: “with limits of detection as low as 0.25 μg/kg (25 ppt), and so high as 0.75 μg/kg (750 ppt).”)

Byron Puga Torres

Reviewer 3 Report

Good work in assessing AFM1 prevalence across provinces of Ecuador. 

I have the following concerns, which you might want to address to improve the paper:

Results section can be better if it is re-written. It does not identify key findings, based on analysis to answer the research questions. This section should be more focused - refer to the alignment of research questions.

On reporting of the results, there is a problem in that you do not seem to point at a finding, followed by a statistical test support. In some cases, the statistical test value begins in a sentence (which is very confusing). For example, you can say ...The level of AFM1 contamination in milk differed among the provinces (P value). Province A had three times more contamination compared to province C. You do not start with ...the Pvalues showed that provinces differed... that sounds confusing. Similarly, you cannot begin by listing tables or figures e.g., Table 1 and 2 show the levels of contamination. That sentence is confusing..

For the discussion section: The first paragraph introduces your work at a higher level to show the strength of the work and then to acknowledge the scope. Here, you show what key contribution the study provides. In subsequent paragraphs, you identify about four major findings that should be discussed as individual paragraphs - some findings can be combined into a single paragraph.

Would it be possible to check whether different types of feed or feeding methods influenced the level of contamination? You might have collected this data in a survey.

Author Response

Response to Reviewer 3 Comments

Point 1: English language and style “Moderate English changes required”

Response 1: We have improved writing in English with the help of a native speaker of the language.

Point 2: Results section can be better if it is re-written. It does not identify key findings, based on analysis to answer the research questions. This section should be more focused - refer to the alignment of research questions.

Response 2: The results section was restructured. The investigation question is answered between lines 120 to 121.

Point 3: On reporting of the results, there is a problem in that you do not seem to point at a finding, followed by a statistical test support. In some cases, the statistical test value begins in a sentence (which is very confusing). For example, you can say ...The level of AFM1 contamination in milk differed among the provinces (P value). Province A had three times more contamination compared to province C. You do not start with ...the Pvalues showed that provinces differed... that sounds confusing. Similarly, you cannot begin by listing tables or figures e.g., Table 1 and 2 show the levels of contamination. That sentence is confusing.

Response 3: Everything requested in the results section was corrected.

Point 4: For the discussion section: The first paragraph introduces your work at a higher level to show the strength of the work and then to acknowledge the scope. Here, you show what key contribution the study provides. In subsequent paragraphs, you identify about four major findings that should be discussed as individual paragraphs - some findings can be combined into a single paragraph.

Response 4: Fixed, some paragraphs have been grouped.

Point 5: Would it be possible to check whether different types of feed or feeding methods influenced the level of contamination? You might have collected this data in a survey.

Response 5: No mycotoxin analysis was performed on dairy cow feed; we know that the food that the animals consumed in the last week before sampling, the same ones that were added in the results section, on lines 122 to 124.

Byron Puga Torres

Round 2

Reviewer 2 Report

Manuscript, after revision, is more understandable and easier to read. Authors should rewrite the conclusions, which at present are a description of the experimental plan used and the factors considered. The authors, in the conclusions, should briefly describe the most important results obtained, i.e. outline the factors that have significantly influenced the contamination of aflatoxins in milk, emphasizing that the areas where the levels are higher, correspond to those where milk production is higher, so more monitoring and surveillance should be necessary. In addition, as a minor revision, the authors, at lines 89-91 should explain whether the contamination data reported for the feeds are bibliographic data, in which case they should indicate references and specify, or are experimental data, in which case they should integrate the sampling method and analytical methods used for the determination of AFLB1 into the materials and methods.

Author Response

Response to Reviewer 2 Comments

Point 1: Manuscript, after revision, is more understandable and easier to read. Authors should rewrite the conclusions, which at present are a description of the experimental plan used and the factors considered. The authors, in the conclusions, should briefly describe the most important results obtained, i.e. outline the factors that have significantly influenced the contamination of aflatoxins in milk, emphasizing that the areas where the levels are higher, correspond to those where milk production is higher, so more monitoring and surveillance should be necessary.

Response 1: The conclusions have been improved, with the indicated suggestions.

Point 2: In addition, as a minor revision, the authors, at lines 89-91 should explain whether the contamination data reported for the feeds are bibliographic data, in which case they should indicate references and specify, or are experimental data, in which case they should integrate the sampling method and analytical methods used for the determination of AFLB1 into the materials and methods.

Response 2: The presence of AFB1 in the food was not analysed, only a feeding survey was carried out that the animals received days before the abalysis of AFM1 in milk. It is clarified on line 110-114.

Reviewer 3 Report

Authors:

Contribution: It is not clear whether the 203 samples represented 30% of milk. Please rephrase.

Please make the results and discussion sections more attractive to the reader, based on the research questions. In the current version, there is more of statistical test values the a reader friendly language. Results: For example, "Analysis of milk showed that province A had three times the amount of AFM1 compared to province B. Interestingly, Province C did not have detectable AFM1 yet it neighbors province A (Table x)

Discussion: Example

Paragraph 1: show overall contribution of this effort and indicate the weakness of the design.

For other paragraphs, highlight key findings and make an opening statement for it in beginning e.g.,

Provinces differed in AFM1 contamination (opening statement)... Then explain what policy or geographical aspects could have contributed. You may use the facts on magnitude differences to further support the claim.

Author Response

Response to Reviewer 3 Comments

Point 1: Contribution: It is not clear whether the 203 samples represented 30% of milk. Please rephrase.

Response 1: Contribution was corrected.

Point 2: Please make the results and discussion sections more attractive to the reader, based on the research questions. In the current version, there is more of statistical test values the a reader friendly language. Results: For example, "Analysis of milk showed that province A had three times the amount of AFM1 compared to province B. Interestingly, Province C did not have detectable AFM1 yet it neighbors province A (Table x).

Discussion: Example

Paragraph 1: show overall contribution of this effort and indicate the weakness of the design.

For other paragraphs, highlight key findings and make an opening statement for it in beginning e.g.,

Provinces differed in AFM1 contamination (opening statement)... Then explain what policy or geographical aspects could have contributed. You may use the facts on magnitude differences to further support the claim.

Response 2: The requested was corrected.

Round 3

Reviewer 3 Report

Results:

Instead of writing, "a p-value of XXX was observed or recorded, begin the sentence as follows, " Significant differences were observed between A and B (P=xx). Move the implications of this statement to the discussion section.

Discussion

Begin with an overview paragraph, to show scope of your work and importance. Don't start with ....Several authors.....NO!

Author Response

Point 1: Instead of writing, "a p-value of XXX was observed or recorded, begin the sentence as follows, " Significant differences were observed between A and B (P=xx). Move the implications of this statement to the discussion section.

Response 1: The requested was corrected (Lines: 79-80; 89-91; 97-98; 163-164; 175; 179).

Point 2: Discussion: Begin with an overview paragraph, to show scope of your work and importance. Don't start with ....Several authors.....NO!

Response 2: The requested was corrected (Lines: 120-125).

Byron Puga Torres
